# Ultrasonic Through-Metal Communication Based on Deep-Learning-Assisted Echo Cancellation

**DOI:** 10.3390/s24072141

**Published:** 2024-03-27

**Authors:** Jinya Zhang, Min Jiang, Jingyi Zhang, Mengchen Gu, Ziping Cao

**Affiliations:** 1Engineering Training Center, Nanjing University of Posts and Telecommunications, Nanjing 210003, China; zhangjy77@njupt.edu.cn; 2College of Telecommunications & Information Engineering, Nanjing University of Posts and Telecommunications, Nanjing 210003, China; 1221014208@njupt.edu.cn (M.J.); 1221014210@njupt.edu.cn (J.Z.); 1022010203@njupt.edu.cn (M.G.)

**Keywords:** echo cancellation, deep-learning, ultrasonic communication

## Abstract

Ultrasound is extremely efficient for wireless signal transmission through metal barriers due to no limit of the Faraday shielding effect. Echoing in the ultrasonic channel is one of the most challenging obstacles to performing high-quality communication, which is generally coped with by using a channel equalizer or pre-distorting filter. In this study, a deep learning algorithm called a dual-path recurrent neural network (DPRNN) was investigated for echo cancellation in an ultrasonic through-metal communication system. The actual system was constructed based on the combination of software and hardware, consisting of a pair of ultrasonic transducers, an FPGA module, some lab-made circuits, etc. The approach of DPRNN echo cancellation was applied to signals with a different signal-to-noise ratio (SNR) at a 2 Mbps transmission rate, achieving higher than 20 dB SNR improvement for all situations. Furthermore, this approach was successfully used for image transmission through a 50 mm thick aluminum plate, exhibiting a 24.8 dB peak-signal-to-noise ratio (PSNR) and a about 95% structural similarity index measure (SSIM). Additionally, compared with three other echo cancellation methods—LMS, RLS and PNLMS—DPRNN has demonstrated higher efficiency. All those results firmly validate that the DPRNN algorithm is a powerful tool to conduct echo cancellation and enhance the performance of ultrasonic through-metal transmission.

## 1. Introduction

In many sensing and control scenarios, signal transmission through metal barriers is necessary without physically penetrating them as much as possible. This includes monitoring the internal running status of a nuclear reactor [1,2], acquiring information from sensors deployed in high-pressure pipes [3,4], and passing data from one side of a watertight bulkhead to the other [5]. Traditional wireless communication based on electromagnetic waves is ineffective due to the Faraday shielding effect. Ultrasound propagates readily in metal; therefore, it has been suggested to replace electromagnetic waves for through-metal communication [6,7,8,9]. However, the acoustic impedance mismatch between the ultrasonic transducer and the metal structure leads to numerous acoustic reflections at interfaces. Consequently, the communication channel presents severe echoing, which further causes intersymbol interference (ISI) at a high data rate and limits the communication rate.

The echoing in the ultrasonic channel is considerably similar to the multipath effect and frequency-selective fading phenomenon observed in radio frequency (RF) communications, which is typically addressed by an equalizer performed at the output of the channel. Some filtering algorithms including least mean square (LMS), normalize least mean square (NLMS) recursive least square(RLS) [10,11,12], subband adaptive filter (SAF), and sparse filter (SF) [13,14,15] have been applied for equalization in the acoustic channel. It was found that frequently-used LMS and NLMS could not satisfyingly remove the echoing impact except its simplicity [10,11]. RLS, SAF and SF exhibited better performance, but their operation complexity partially prevented the actual utilization [13,14,15,16].

The echo cancellation was also investigated by using an experimental-based or model-driven filter at the input of the channel to pre-distort signals, which might simplify the receiver’s duties [17,18,19]. Primerano et al. introduced a canceling pulse at the transmitter of their communication system and destructively combined it with the echoes to suppress its amplitude [17]. Their report demonstrated that the data rate was significantly enhanced from 50 kbps to 1 Mbps. Following that, they further improved the echo cancelation process by building a more accurate model and using that model to derive a better pre-distorting filter [18]. Actually, the pre-distorting filter was constructed based on certain channel conditions, for instance, the metal type and wall thickness, and hence it easily lost robustness due to the diversity and variability of practical applications.

On the other hand, orthogonal frequency division multiplexing (OFDM) is considered a powerful solution to the multi-path problem. The OFDM technique divides the frequency-selective channel into orthogonal flat fading bands, allowing for a marked increase in data rate without using highly complex equalizers to correct signal fading. Previous literature indicated that OFDM-based communications could accomplish high-data-rate transmission through metal barriers [20,21,22]; however, one shortcoming of OFDM is that it will result in a high peak-to-average power ratio (PAPR). High PAPR means that the system should consume high power and employ a power amplifier with a broad linear range for the transmitter.

Over the past decade, deep learning (DL) has been widely used in engineering fields [23,24], for example, natural language processing. Some DL algorithms have been proven to be greatly efficient for treating unintelligible subjects in the physical layer of wireless communication where the channel conditions are unknown or too complicated for analyzing [25,26,27]. Recent reports indicated that DL-based methods were substantially powerful for echo cancellation at audio frequency, implying its potential application in ultrasonic communication [28,29]. It is noted that the ultrasonic wave used for through-metal communication is generally in the range of hundreds of kHz, which is two to three orders of magnitude higher than audio frequency. In this paper a DL algorithm of dual-path recurrent neural network (DPRNN) was investigated to undertake this task for an ultrasonic through-metal communication system. Additionally, to evaluate the validity of the DPRNN algorithm, the conventional LMS approach is comparatively employed to equalize the same channel. The communication system assisted with the trained DPRNN model exhibited a higher signal-to-noise ratio (SNR) signal transmission than the one only using LMS. Finally, the transmission of an image with 100 × 100 pixels was achieved in a 50 mm long aluminum channel to verify the advantage of the DPRNN strategy, and a 24.8 dB peak-signal-to-noise ratio (PSNR) and 95% structural similarity (SSIM) were, respectively, obtained for the recovered image.

## 2. Materials and Methods

### 2.1. Echoing Problem

As shown in Figure 1, the ultrasonic transmission channel consisted of a pair of coaxially aligned transducers and a 50 mm thick aluminum plate. The nominal resonance frequency of both transducers was 10 MHz. To obtain a good coupling, a thin layer of couplant gel was added between each one of the transducers and the metal plate. During propagation, ultrasound experiences reflection and refraction at the inner surface of the metal plate, causing the carried signals to reach the receiver through multiple paths. Due to the different lengths of these paths, the signals in the receiver exhibit various delays and attenuations, which further leads to the incorrect superposition of multiple signals. Consequently, signal distortion occurs and the communication performance deteriorates.

An impulse response test for this channel was conducted with a signal generator (DG1022U, RIGOL, Beijing, China) and an oscilloscope (DSO-X 2012A, Agilent, Beijing, China). In this test, the transmitting transducer was excited with a pulse of 20 volts and 50 ns (this pulse width of 50 ns was equal to one-half of the resonant period of transducers being chosen for maximizing the response amplitude of the receiving transducer), and the response at the receiving transducer was recorded with the oscilloscope. Figure 2 plotted the data collected from the impulse response experiment. The trace in Figure 2a is the signal exerted on the transmitting transducer, and the trace in Figure 2b is the waveform collected at the receiving transducer. It was found that the transducer acted as a bandpass filter, resonating at approximately 10 MHz. According to Figure 2, it can be calculated that the signal was attenuated by 17.96 dB by the time it reached the receiver.

The ultrasonic channel in Figure 1 could be looked at as a two-port network with transmitted and reflected power at each port. For obtaining the channel characteristics, a vector network analyzer (E5080A, KEYSIGHT, Beijing, China) was used to measure its scattering parameters which included the reflection coefficient for the transmitted port $S_{11}$, the reverse power gain $S_{12}$, the forward power gain $S_{21}$, and the reflection coefficient for the received port $S_{22}$. The voltage transfer function H(f) of the channel was calculated with these parameters using the expression:(1)$$H\left(f\right)=\frac{S_{21}\left(f\right)}{1+S_{11}\left(f\right)}$$
(2)$$\begin{array}{c} \left[\begin{array}{c} b_{1}\\ b_{2} \end{array}\right]=\left[S\right]\times \left[\begin{array}{c} a_{1}\\ a_{2} \end{array}\right]=\left[\begin{array}{cc} S_{11} & S_{12}\\ S_{21} & S_{22} \end{array}\right]\times \left[\begin{array}{c} a_{1}\\ a_{2} \end{array}\right] \end{array}$$
where $a_{1}$ and $a_{2}$ were incident power waves on port 1 and port 2, respectively, and $b_{1}$ and $b_{2}$ were reflected power waves from port 1 and port 2, respectively.

Figure 3 shows a plot of ${\left|S_{21}\right|}^{2}$ ($20{log}_{10}\left|S_{21}\right|$(dB)), the magnitude response of the channel. A dramatic frequency selectivity with periodic channel fading can be observed, which is attributed to the echoing of acoustic energy in the channel. For a frequency-selective channel, the multipath delay spread is defined as the arrival time difference between the first significant multipath and the last multipath, and its coherence bandwidth is defined as the frequency bandwidth over which the channel is correlated. These two indexes have a relationship as
(3)$$F_{c}=\frac{1}{D}$$
where *D* is the delay spread, and $F_{c}$ is the coherence bandwidth.

The impulse response in Figure 2 clearly indicates that the first multipath signal in the channel appeared at 7.97 μs which was approximately the time it took for the ultrasound to travel through the 50 mm thick aluminum barrier. The speed of sound in aluminum is 6.27 km/s. The last detectable multipath signal arrived at 57.06 μs, meaning that the delay spread *D* of the channel was approximately 49.09 μs. According to Equation (Equation 3), $F_{c}$ should be 20.37 kHz. Therefore, when a signal with a bandwidth larger than 20.37 kHz passes through the channel, severe intersymbol interference will appear. The intersymbol interference problem in RF communications is generally dealt with by the channel equalization method. Taking into account that channel equalization was considerably complex for an uncertain or unstable channel, in this study an elaborately trained DPRNN model was introduced into the communication system to perform the echo cancellation and hinder intersymbol interference.

### 2.2. System Architecture

Figure 4 depicts a block diagram of an ultrasonic through-metal communication system assisted with two alternative approaches of echo cancellation. The upper part of this figure is the system architecture similar to a general one, comprised of a signal transmitting side, a 50 mm long metal channel, and a signal receiving side. Signals undergo several processing steps upon entering the transmitting side, including source coding, ASK digital modulation, DA conversion, and signal amplification. Following that, the modulated ultrasound is transmitted into the metal channel via electroacoustic conversion afforded by the transmitting transducer. On the signal receiving side, ultrasonic signals are subjected to the reverse process including acoustoelectric conversion, signal amplification, AD conversion, synchronization, channel equalization, digital demodulation, and source decoding.

The lower part of Figure 4 illustrates that two alternative routes, LMS adaptive filtering and DPRNN processing, are utilized to replace the equalization module in the above system. LMS adaptive filtering as a typical equalization approach has been frequently used in traditional communication systems, here being conducted just for making a comparison of equalization efficiency with DPRNN. As for the DPRNN route, it simultaneously accomplishes equalization and synchronization tasks, which are mainly composed of segmentation, block processing and overlap-add. The introduction of LMS adaptive filtering or DPRNN algorithm into the channel mitigates the intersymbol interference that resulted from acoustic echoing.

### 2.3. Echo Cancellation Approach I: Adaptive Filtering

The internal structure of the LMS adaptive filter is shown in Figure 5, and adaptive algorithms are used to update the filter coefficients of the adaptive filter. LMS is a search algorithm in which weight coefficients $w\left(n\right)$ of a filter are updated on the basis of the steepest descent, moving weight coefficients in proportion to the instantaneous gradient estimate of the mean square error [30]. The formula is as follows: (4)$$y\left(n\right)=w^{T}\left(n\right)x_{N}\left(n\right)$$
(5)$$e\left(n\right)=d\left(n\right)-y\left(n\right)$$
(6)$$w\left(n+1\right)=w\left(n\right)+2\mu e\left(n\right)x_{N}\left(n\right)$$
where $y\left(n\right)$, ${\left(\;\cdot \;\right)}^{T}$, $x^{T}\left(n\right)$, $e\left(n\right)$, $d\left(n\right)$ and μ are the output signal, transposition operation, input signal, error signal, reference signal and step size, respectively. *N* denotes the filter order, which means the number of delay units in the filter, or the dimension of the filter’s weight vector. It can be seen that the filter coefficient update is driven by the error signal which generates filter coefficients in a manner of minimizing the error signal.

The PNLMS adaptive filter uses a variable step size parameter proportional to the filter taps’ sparsity to adjust the convergence rate. It determines the active state based on tap sparsity, assigning larger step sizes to active taps for faster convergence and smaller step sizes to inactive taps for reduced steady-state error. Literature indicated that PNLMS had a much higher convergence rate than LMS. The PNLMS filter weight update formula is as follows: (7)$$w\left(n+1\right)=w\left(n\right)+\frac{\mu G(n+1)}{\delta +x^{T}\left(n\right)G\left(n+1\right)x\left(n\right)}\;\cdot \;e\left(n\right)x\left(n\right)$$
where G(n) is an N-dimensional diagonal gain matrix used to allocate step length weights to the filter’s weight coefficients, and δ is a regularization parameter.

The fundamental principle of the RLS algorithm involves utilizing the available observation data to estimate the necessary filter coefficients and minimizing the mean square error of prediction in order to optimize the filtering effect. The formula is as follows: (8)w(n)=w(n−1)+k(n)e(n)
(9)$$k\left(n\right)=\frac{P(n-1)x(n)}{\lambda +x^{T}\left(n\right)P\left(n-1\right)x\left(n\right)}$$
(10)$$P\left(n\right)=\frac{1}{\lambda }\left[P\left(n-1\right)-k\left(n\right)x^{T}\left(n\right)P\left(n-1\right)\right]$$
where k(n), P(n) and λ are the gain vector, inverse matrix and forgetting factor. The further away from time n, the smaller the weight allocated to the error, with λ determining the magnitude of the weight between 0 and 1.

### 2.4. Echo Cancellation Approach II: DPRNN

In comparison to traditional recurrent neural networks (RNN), DPRNN is a recently proposed neural network algorithm with much higher computation efficiency for long sequence modeling [31,32,33,34]. RNN readily suffers from gradient vanishing and explosion when handling long sequential data and exhibits long-term dependency. DPRNN handles long sequences in a unique way of segmenting them into shorter sub-sequences of fixed length. This network structure incorporates two types of RNN: an intra-chunk RNN in which the training within sub-sequence is conducted, and an inter-chunk RNN in which the modeling operation between sub-sequences is conducted, respectively, responsible for handling local and global information. Each RNN block receives only a small number of time steps throughout the entire sequence and thereby simplifies the optimization training. Considering that ultrasonic communication involves the transmission of long sequential signals, we pioneeringly take advantage of DPRNN as a reasonable approach for echo cancellation.

As Figure 6 shows, the DPRNN architecture is comprised of three stages: segmentation, block processing, and overlap-add. In the first stage, the continuous input is divided into overlapped blocks, and then all these blocks are concatenated into a three-dimensional tensor. Secondly, the tensor is passed to stacked DPRNN blocks to iteratively perform local and global modeling. Finally, the output of the last layer is transformed into an output sequence through the overlap-add way.

The segmentation operation divides continuous input signals with a feature dimension of *N* and a time step of *L* into overlapped blocks of length *K* and hop size *P*, where K=2P. This performance results in S blocks with equal size, denoted as $D_{s}\in R^{N\times K},\;s=1,\ldots ,S$. Subsequently, all the blocks are concatenated to form a 3D tensor $T=\left[D_{1},\ldots ,D_{S}\right]\in R^{N\times K\times S}$.

Block processing takes the segmented output *T* and passes it through *B* DPRNN blocks. Each block accomplishes both intra- and inter-chunk processing, yielding another $T_{b}\in R^{N\times K\times S}$ with the same shape as the input, where b=1,…,B. The intra-clunk RNN is kept bidirectional, and applied to the second dimension of $T_{b}$: (11)$$U_{b}=f_{b}\left(T_{b,s}\right)$$

Here, $U_{b}\in R^{H\times K\times S}$ is the output of the intra-chunk RNN, H is the hidden dimension of the RNN, and $f_{b}\left(\;\cdot \;\right)$ is the mapping function. $T_{b,s}=T_{b}\left[:,:,s\right]\in R^{N\times K},\;s=1,\ldots ,S$ denotes the sequence corresponding of data block *s*. The feature dimension of $U_{b}$ is transformed to match that of $T_{b}$ using a linear fully-connected (FC) layer, and then layer normalization (LN) is conducted to enhance generalization: (12)$$LN\left({\hat{U}}_{b}\right)=\frac{{\hat{U}}_{b}-\mu \left({\hat{U}}_{b}\right)}{\sqrt{\sigma \left({\hat{U}}_{b}\right)+\varepsilon }}⊙z+r$$
where ${\hat{U}}_{b}=\left(GU_{b,s}+m\right)$ is the transformed output, $G\in R^{N\times H}$ and $m\in R^{N\times 1}$ are, respectively, the weights and biases of the FC layer, *z* and $r\in R^{N\times 1}$ are scaling factors, μ(·) and σ(·) represent, respectively, the mean and variance of the 3D tensor, and ⊙ denotes the Hadamard product. A residual connection is established by adding the LN output to its input, formulated as ${\hat{T}}_{b}=T_{b}+LN({\hat{U}}_{b})$. The intermediate tensor ${\hat{T}}_{b}$ is then fed as the input to the inter-chunk RNN. This feeding operation is conducted on its third dimension which corresponds to the *k* time steps within each block: (13)$$V_{b}=h_{b}\left({\hat{T}}_{b,k}\right)$$

Here, $V_{b}\in R^{H\times K\times S}$ denotes the output of the RNN, $h_{b}\left(\;\cdot \;\right)$ is the mapping function defined by the RNN and ${\hat{T}}_{b,k}$ is the sequence corresponding to the *k*-th time step within each block. Similar to the inter-chunk RNN, $V_{b}$ also undergoes linear FC layer and then LN operations, being further treated with the addition of a residual connection between the output and ${\hat{T}}_{b}$ to form the output of block *b*. For b<B, the output of block *b* acts as the input to the next block $T_{b+1}$.

Finally, the *S* blocks are recombined using an overlap-add method to obtain the final output $Q\in R^{N\times L}$, meaning that the reconstruction of the long sequence signal has been accomplished.

### 2.5. Implementation of Signal Transmission

Each module in the aforementioned communication system was actually constructed with the means of hardware or software for the signal transmission experiment. Figure 7 illustrates the constructed signal transmitting side, which utilizes a field programmable logic gate array (FPGA, EP4CE10F17C8, Cyclone IV E) as its core unit. The FPGA contains two image acquisition methods: camera and data interface, and generates a carrier wave with its numerically controlled oscillator (NCO). After image data are collected by the FPGA, they undergo sequentially SDRAM buffering, parallel-serial conversion, and ASK digital modulation. Since the output signals of the FPGA are digital, a DA circuit is applied to carry out the digital-to-analog conversion. Converted signals further experience bandpass filtering to remove out-of-band noise and their amplitude is increased by an amplification circuit. Finally, the ultrasonic transmitting transducer is driven to transmit ultrasonic signals into the metal channel.

As depicted in Figure 8, the signal-receiving side is actually constructed with an amplification circuit, an AD conversion circuit, and a personal computer (PC). After the received signals pass through amplification and AD conversion circuits, they are captured by an oscilloscope and then sent to the PC. Within the PC, signals are treated with two routes of echo cancellation. One route is called LMS adaptive filtering which is built with Matlab. Another one is DPRNN processing which is established with Python. The implementation of ASK demodulation, serial-parallel conversion and image restoration are also performed on the PC.

A photo of the constructed ultrasonic communication system is shown in Figure 9. It is noted that data were simultaneously collected at two sites of the transmitting transducer and AD conversion circuit with the oscilloscope, denoted as $S_{T}\left(t\right)$ and $S_{R}\left(t\right)$, respectively. The carrier frequency $f_{c}$ was set as 10 MHz, while the sampling frequency $f_{s}$ was set as 50 MHz, and the transmission rate was kept at 2 Mpbs. Continuous data sampling was conducted for a duration of 1.2 s, producing 60 million data points at each signal collection site. As for the DPRNN route of echo cancellation, the data streams $S_{T}\left(t\right)$ and $S_{R}\left(t\right)$ were, respectively, partitioned into 20,000 segments, with each segment containing 3000 data points. By grouping together corresponding data segments from $S_{T}\left(t\right)$ and $S_{R}\left(t\right)$ that were captured simultaneously, a total of 20,000 data sets were formed.

The training of the LMS adaptive filter and DPRNN model was conducted on a PC equipped with a Geforce MX450 (2G) graphics card. The DPRNN model was developed based on the PyTorch deep learning framework, utilizing Python interpreter version 3.8. Signals captured from the transmitting transducer served as source signals, and were used as the labels for the model; signals captured from the receiving transducer contained echoes and noise. The data treated with echo cancellation of LMS and DPRNN were signed as $S_{LMS}\left(t\right)$ and $S_{DPRNN}\left(t\right)$, respectively.

## 3. Results

### 3.1. Data Transmission

#### 3.1.1. Generating Initial Received Signals with Different SNRs

In ultrasonic communication systems, the characteristics of the channel are highly sensitive to many physical conditions such as channel length, the coaxial alignment offset of the transmitting and receiving transducers, and environmental temperature. Any variation in channel conditions may lead to fluctuations in energy dissipation, acoustic echo, and noise intrusion within the channel. These variations and fluctuations generally have a considerable impact on signal attenuation, delay, and waveform distortion. SNR is one of the most crucial metrics for assessing signal quality. In order to evaluate the effectiveness of the LMS adaptive filter and DPRNN models for echo cancellation, we artificially made the received signals $S_{R}\left(t\right)$ with different SNRs. Here, SNR was defined as the ratio of the power of the reference waveform to the power of the estimated waveform as follows: (14)$$SNR\left(x,\hat{x}\right)=10log_{10}\frac{{∥\begin{array}{c} \hat{x} \end{array}∥}^{2}}{{∥\begin{array}{c} \hat{x}-x \end{array}∥}^{2}}$$
where *x* and $\hat{x}$ represented the estimated waveform and reference waveform, respectively.

Two methods were adopted to generate the above-mentioned signals at a 2 Mbps transmission rate. Method 1 was to change the channel length by using aluminum plates with thicknesses of 30 mm, 40 mm, 50 mm, and 60 mm, respectively. Method 2 was to adjust the alignment offset distance of the transmitting and receiving transducers along their axis while keeping the 50 mm long aluminum channel for data transmission. For both transducers with the same diameter of 16 mm, offset distances were set as 0 mm, 6 mm, 12 mm, and 18 mm, respectively. Table 1 shows the different SNRs of the signals collected at the AD conversion circuit of the signal receiving side under various channel conditions, implying that SNR is highly affected by channel conditions.

#### 3.1.2. Quality Evaluation Indexes of Signal Recovery

Besides SNR, we introduced scale-invariant signal-to-noise ratio (SISNR), normalized correlation coefficient (NCC), bit error rate (BER) and echo return loss enhancement (ERLE), to more sufficiently assess the applicability of the LMS adaptive filter and DPRNN model for echo cancellation. SISNR might remodify the clean signal using a scaling factor, ensuring that the distorted portion and the clean signal vector are orthogonal. This index eliminates substantial distortion resulting from the SNR when the generated and true signals hold parallel directions and different amplitudes but share similar waveforms. SISNR is formulated as below: (15)$$SISNR=10\log_{10}\frac{\left|\right|x_{target}{\left|\right|}^{2}}{\left|\right|e_{noise}{\left|\right|}^{2}}$$
(16)$$x_{target}=\frac{\left(\hat{x},x\right)x}{{\left||x|\right|}^{2}}e_{noise}=\hat{x}-x_{target}$$
where, $x_{target}$ denotes the norm of the clean signal, and the $e_{noise}$ represents the norm of the difference between the clean signal and the estimated signal.

NCC is a parameter to quantify the overall waveform similarity before and after signal denoising, which is expressed as follows: (17)$$NCC=\frac{\sum_{n=1}^{N}A_{s}\left(n\right)A_{d}\left(n\right)}{\sqrt{(\sum_{n=1}^{N}\overset{2}{\;A_{s}\left(n\right)})(\left(\sum_{n=1}^{N}\overset{2}{\;A_{d}\left(n\right)})\right)}}$$
where $A_{s}\left(n\right)$ is the reference signal and $A_{d}\left(n\right)$ is the separated signal. In this study, a higher NCC means a better efficiency of echo cancellation.

BER is the probability of code transmission error and is calculated as: (18)$$P_{e}=\frac{1}{4}erfc\left(\sqrt{\frac{r}{4}}\right)+\frac{1}{2}e^{-\frac{r}{4}}$$
where $r=20\mathrm{lg}(V_{s}/V_{n})$, $V_{n}$ is the amplitude of received signals, and $V_{n}$ represents the amplitude of the noise in received signals.

ERLE is a metric used to evaluate the performance of echo cancellation in acoustic systems, indicating the degree of improvement in the audio signal after echo elimination. ERLE is defined as below: (19)$$ERLE=10\log_{10}\left(\frac{E\{y^{2}\left(n\right)\}}{E\{s^{2}\left(n\right)\}}\right)$$

In the equation, y(n) represents the desired signal and s(n) represents the estimated signal. A higher ERLE value indicates a better echo cancellation.

#### 3.1.3. Comparison of Echo Cancellation Performance

During the training of the DPRNN model, the dataset of 20,000 pairs of data was divided into a training set, validation set, and test sets with a ratio of 6:2:2, ensuring sizable validation and test sets for evaluating the model performance. Considering the computational efficiency and memory capacity limitation, the block size K for segmentation was set as 250. Within each block, a bidirectional long-short-term memory (LSTM) layer [35] was employed with 128 hidden units in each direction to effectively capture complex temporal dependencies in both directions. The configuration of the inter-block RNN was the same as the intra-block RNN. The DPRNN model consisted of two interleaved intra-block RNNs and two inter-block RNNs. The Adam optimizer [36] was adopted with an initial learning rate of 0.001, a learning rate decay of 0.95 for every two epochs, and an initial training of 50 epochs for all training sets. The decay of the learning rate is used as a rule for fine-tuning the model over the entire duration. This prevents it from getting stuck in suboptimal solutions. To maximize the SISNR value during the training, SISNR was used as the loss function in the DPRNN model.

Figure 10 plots the training loss and valid loss of the trained network with different epochs. It can be observed that the validation set’s loss function began to stabilize around the 40th epoch. To prevent overfitting, the model trained with 40 epochs was selected for echo cancellation in the ultrasonic through-metal communication.

Table 2 lists the signal quality of DPRNN-based echo cancellation, and compares it with the LMS, PNLMS and RLS algorithms. All four evaluation indexes of the signals treated with the DPRNN algorithm are very impressive. In practice, NCC of 0.9829, BER of 3.502 × $10^{-4}$ and ERLE of 32.53 should sufficiently meet most application requirements. Compared with the adaptive algorithms listed, the DPRNN algorithm displayed a more distinct improvement in efficiency, and its BER was two orders of magnitude higher than that of LMS.

Figure 11 presents the partial waveforms of signals before and after LMS, RLS, PNLMS and DPRNN echo cancellation, i.e., six series of signals: $S_{T}\left(t\right)$ and $S_{R}\left(t\right)$ are, respectively, collected on the transmitting and receiving transducer; $S_{LMS}\left(t\right)$, $S_{RLS}\left(t\right)$ and $S_{PNLMS}\left(t\right)$ are independently collected after LMS, RLS and PNLMS adaptive filtering; and $S_{DPRNN}\left(t\right)$ collected after DPRNN treatment. A 50 mm thick aluminum plate and an 18 mm offset distance of axial alignment are chosen as the channel conditions. It can be found that after being processed with the DPRNN model, echoes as well as other noises have been fairly eliminated from the source signals, firmly proving the effectiveness of the DPRNN-based approach. Moreover, the distortion level of signals dealt with the DPRNN approach is far lower than that of signals treated with adaptive filtering.

As shown in Figure 12, SNR of initial received signal apparently affects echo cancellation of LMS, RLS, PNLMS and DPRNN. Low SNR of initial received signal suppresses the efficiency of the adaptive filter and DPRNN model, but compared with adaptive algorithms listed, DPRNN exhibits higher bearing capability against this impact in a fashion of relatively slow declining rate. After these signals with different initial SNRs were conducted with DPRNN echo cancellation, their SNR improvements were still higher than 20 dB, implying that they were greatly applicable.

### 3.2. Image Transmission

Based on the aforementioned adaptive filter and trained DPRNN model, we conducted the image transmission through a 50 mm thick aluminum metal plate. On the transmitting side, the FPGA chip acquires the data of an image with 100 × 100 pixels with one of its I/O interfaces and saves them in its ROM. This image has the RGB565 color format in which each pixel uses 16 bits for color value. As shown in Figure 13, each pixel cell is composed of start, sync, data, parity, end and guard interval bits, entirely occupying 32 bits. The following steps of signal transmission were the same as those of the aforementioned data transmission. The image recovery was also implemented with Matlab, being involved multiple steps: signal reading, AC-DC conversion, envelope extraction, sampling and decision-making. Finally, the recovered images were displayed on the PC screen.

Figure 14 displays images recovered from signals treated with different approaches of echo cancellation. Figure 14b is the recovered image without any echo cancellation, in which severely distorted lines and colors imply the existence of strong intersymbol interference. Figure 14c is the image recovered after received signals undergo LMS adaptive filtering. The distortion degrees of patterns and colors in Figure 14c have been obviously mitigated in comparison with Figure 14b, revealing that LMS adaptive filtering can serve echo cancelation and weaken intersymbol interference to some extent. Figure 14d represents the image recovered after received signals were processed with DPRNN echo cancellation. In this case, no discernible distortion can be found, meaning that both intersymbol interference and noise have been almost eliminated, and DPRNN is significantly effective for echo cancellation.

To quantitatively evaluate the image recovery, two metrics are adopted: peak signal-to-noise ratio (PSNR), structural similarity index measure (SSIM) and color difference value (ΔE). PSNR denotes the ratio of the maximum possible power of a signal to the power of the noise that affects its representation accuracy, formulated as below: (20)$$PSNR=10\mathrm{lg}\left(\frac{{MAX}_{I}^{2}}{MSE}\right)$$
(21)$$MSE=\frac{1}{mn}\sum_{j=0}^{n-1}{\left[I\left(i,j\right)-K\left[i,j\right]\right]}^{2}$$
where $MAX_{I}$ is the possible maximum pixel value in the image. If the pixels are represented by N-bit binary, ${MAX}_{I}=2^{N}-1$. MSE denotes the mean square error, with *I* representing the clean image and *K* representing the noisy image.

SSIM is a reliable scale for measuring the similarity between two images of x and y and contains three major features: luminance, contrast, and structure. A higher SSIM value indicates better image restoration quality in this study. SSIM is calculated as follows: (22)$$SSIM\left(x,y\right)={l\left(x,y\right)}^{\alpha }\;\cdot \;{c\left(x,y\right)}^{\beta }\;\cdot \;{s\left(x,y\right)}^{\gamma }$$
(23)$$l\left(x,y\right)=\frac{2\mu_{x}\mu_{y}+c_{1}}{\mu_{x}^{2}+\mu_{y}^{2}+c_{1}}$$
(24)$$c\left(x,y\right)=\frac{2\sigma_{x}\sigma_{y}+c_{2}}{\sigma_{x}^{2}+\sigma_{y}^{2}+c_{2}}$$
(25)$$s\left(x,y\right)=\frac{\sigma_{xy}+c_{3}}{\sigma_{x}\sigma_{y}+c_{3}}$$

In the SSIM formula, $l\left(x,y\right)$, $c\left(x,y\right)$, and $s\left(x,y\right)$ denote the similarity of luminance, contrast, and structure between *x* and *y*, respectively. $\mu_{x}$ and $\mu_{y}$ are, respectively, the mathematical means of *x* and *y*; $\sigma_{x}^{2}$ and $\sigma_{y}^{2}$ are, respectively, the variances of *x* and *y*, and $\sigma_{xy}$ is the covariance between *x* and *y*. The constants $c_{1}$, $c_{2}$, and $c_{3}$ are used to avoid division by zero. α, β, and γ represent the weighting factors for different features in the SSIM measurement, here customarily set to 1.

ΔE as an important indicator was proposed by the International Commission on Illumination (CIE) to evaluate color accuracy, with the ΔE2000 defined as: (26)$$\Delta E=\sqrt{{\left(\frac{\Delta L^{\prime }}{K_{L}S_{L}}\right)}^{2}+{\left(\frac{\Delta C^{\prime }}{K_{C}S_{C}}\right)}^{2}+{\left(\frac{\Delta H^{\prime }}{K_{H}S_{H}}\right)}^{2}+R_{T}\frac{\Delta C^{\prime }\Delta H^{\prime }}{K_{C}S_{C}K_{H}S_{H}}}$$

The equation includes parameters $K_{L}$, $K_{C}$, $K_{H}$ for luminance, chroma, and hue factors; ΔL, ΔC, ΔH for their respective differences and weighting factors $S_{L}$, $S_{C}$, $S_{H}$ for each. $R_{T}$ is the rotation factor. By calculating the ΔE between the original image and the image restored after echo cancellation, color fidelity can be evaluated. Typically, a smaller ΔE value indicates a higher quality of image restoration.

Table 3 shows the computed evaluation indexes of image restoration for different echo cancellation methods. As can be seen, both the LMS filter and DPRNN model remarkably enhance the image restoration level, and the DPRNN model is far more efficient in echo cancellation than the LMS filter. SSIM in the DPRNN case is up to 0.9471 high, which is quite favorable from the view of image restoration.

## 4. Conclusions

In this study, the application of the DPRNN approach for echo cancellation was investigated in the ultrasonic through-metal communication system. Firstly, the characterization of the ultrasonic channel was examined through the experiment of impulse response and magnitude response, revealing that owing to the echo cancellation, the coherence bandwidth was only 20.37 kHz and strong frequency selectivity existed. Subsequentially, an actual system was constructed based on software and hardware, consisting of a pair of ultrasonic transducers, an FPGA module, some lab-made circuits, etc. Two approaches of LMS and DPRNN echo cancelation were comparatively applied to signals with different SNRs at a 2 Mbps transmission rate. Both exhibited a substantial improvement in signal quality, but the DPRNN one evidently achieved much higher efficiency with 20 dB SNR improvements for all situations. Furthermore, the image transmissions through a 50 mm thick aluminum plate were conducted in which the DPRNN also performed well, and achieved a 24.8 dB PSNR and about 95% SSIM. We sufficiently demonstrated that the DPRNN echo cancellation was a suitable choice to improve the ultrasonic through-metal transmission.

## Figures and Tables

**Figure 1 sensors-24-02141-f001:**
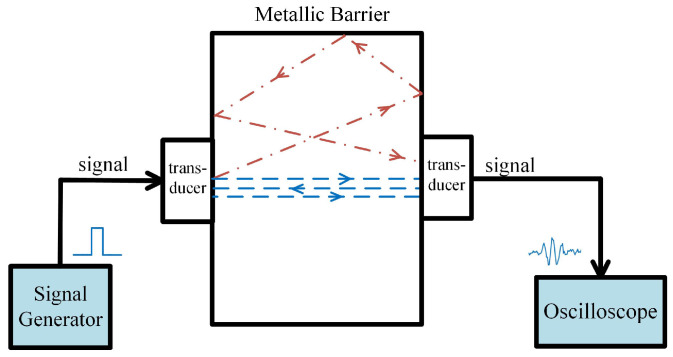
Ultrasonic transmission channel.

**Figure 2 sensors-24-02141-f002:**
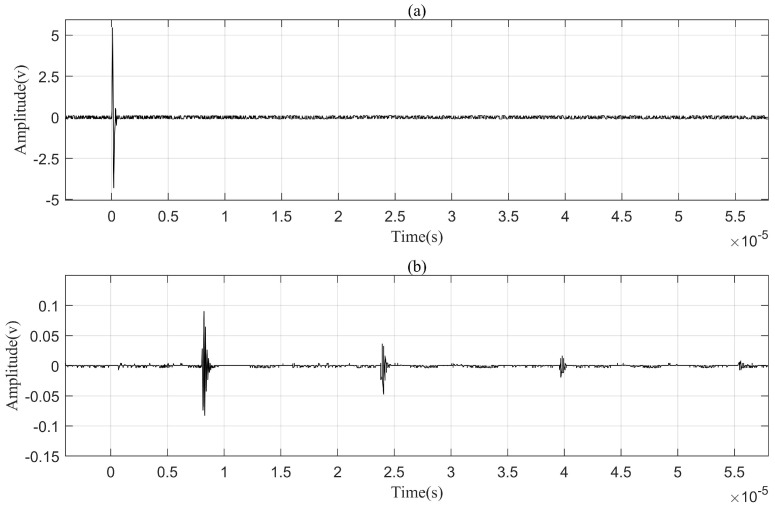
Impulse response of the 50 mm long aluminum channel with a pair of coaxially aligned ultrasonic transducers. (**a**) signal exerted on the transmitting transducer. (**b**) signals collected at the receiving transducer.

**Figure 3 sensors-24-02141-f003:**
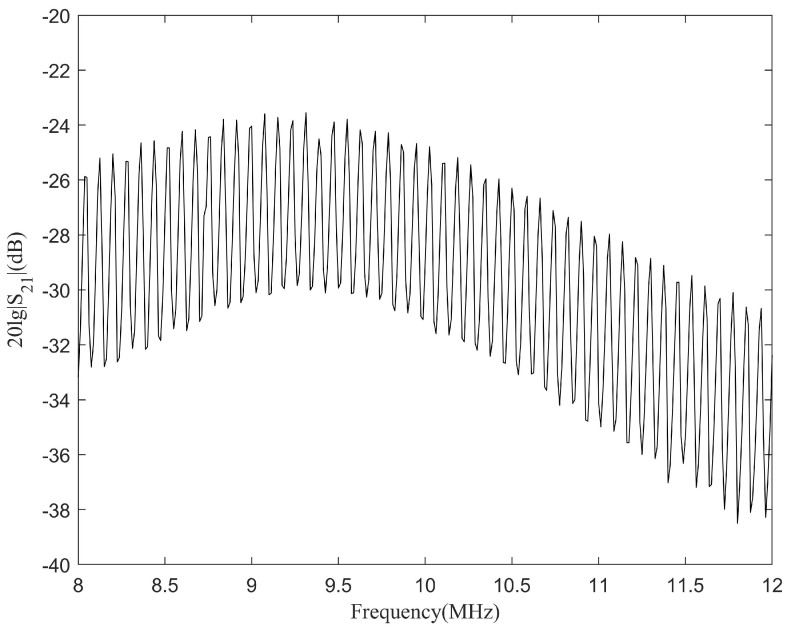
Magnitude response of the 50 mm long aluminum channel with a pair of coaxially aligned ultrasonic transducers.

**Figure 4 sensors-24-02141-f004:**
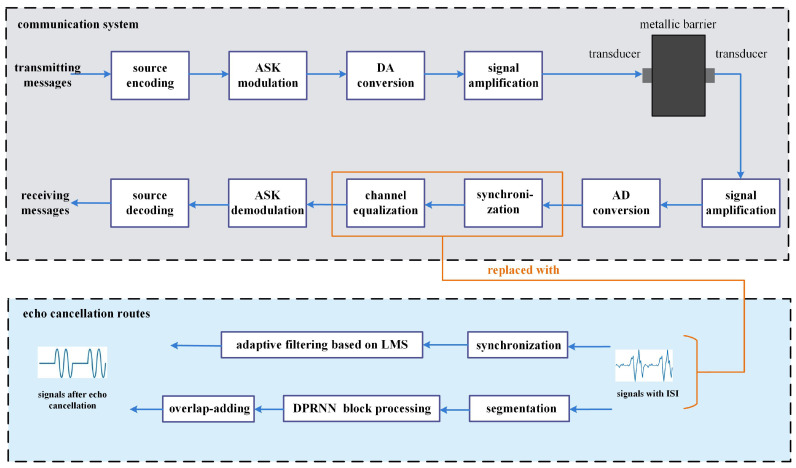
System architecture of ultrasonic through-metal communication.

**Figure 5 sensors-24-02141-f005:**
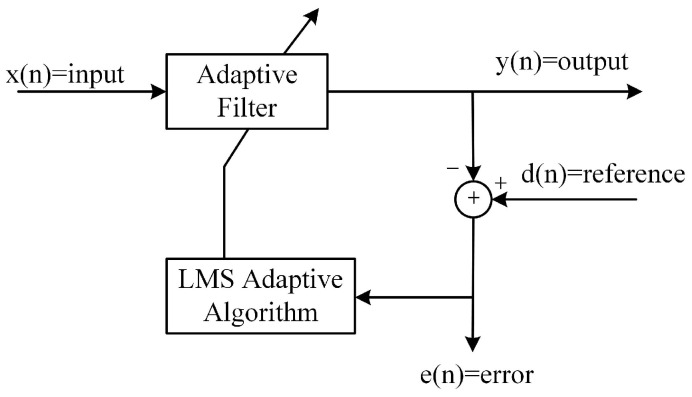
Internal structure of the LMS adaptive filtering.

**Figure 6 sensors-24-02141-f006:**
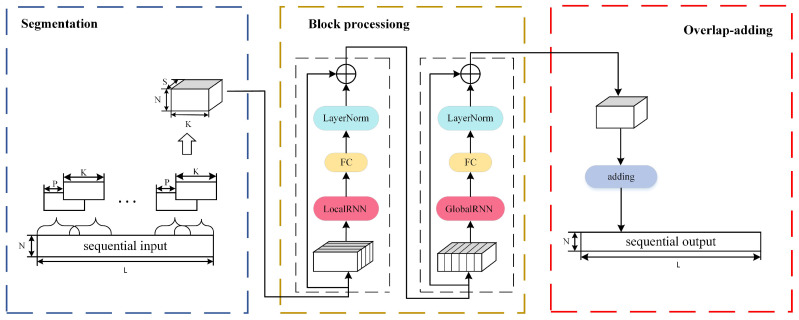
Block diagram of DPRNN algorithm.

**Figure 7 sensors-24-02141-f007:**
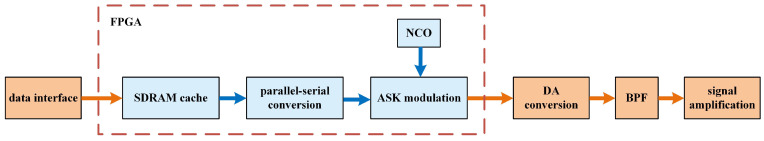
Actual construction of the signal transmitting side.

**Figure 8 sensors-24-02141-f008:**
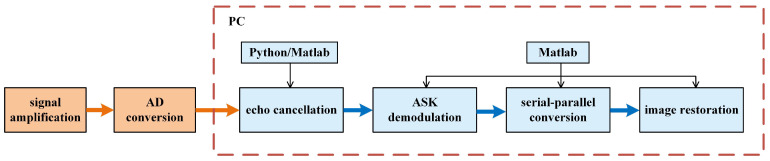
Actual constructure of the signal receiving side.

**Figure 9 sensors-24-02141-f009:**
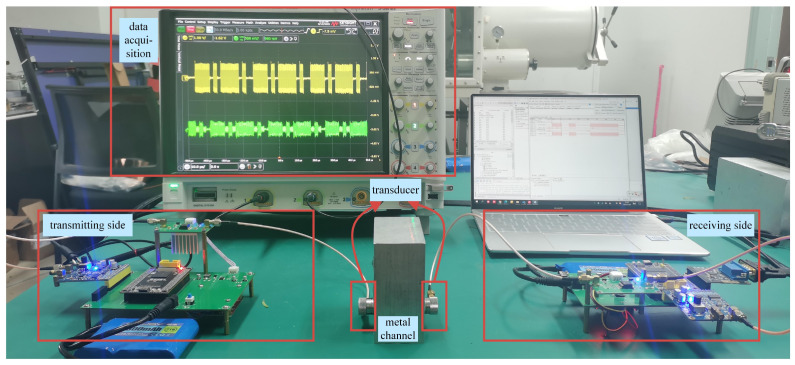
Photo of actual ultrasonic through-metal communication system.

**Figure 10 sensors-24-02141-f010:**
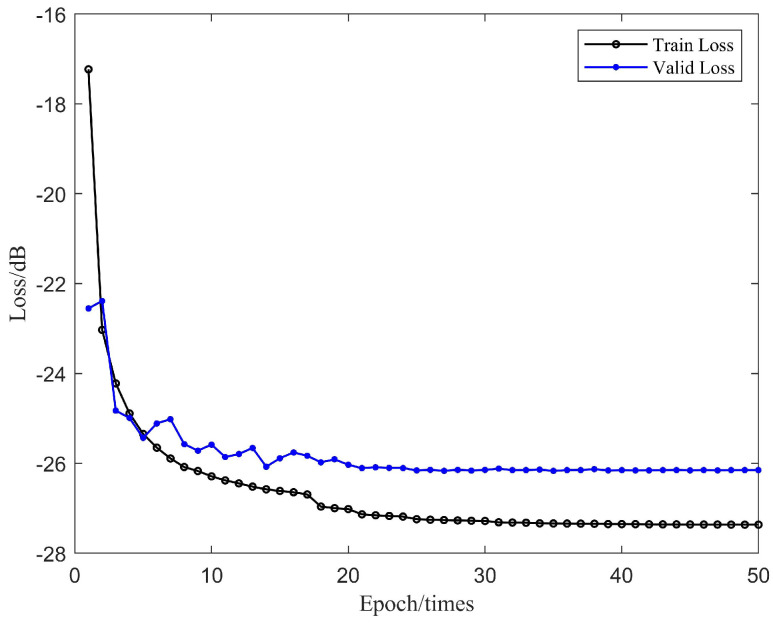
Loss function curves of DPRNN for echo cancellation.

**Figure 11 sensors-24-02141-f011:**
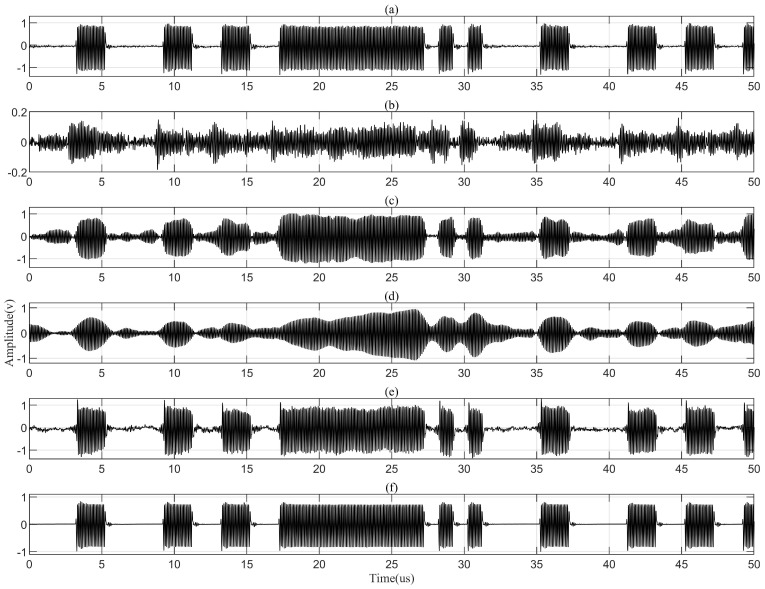
Signals before and afte echo cancellation. (**a**) Signals $S_{T}\left(t\right)$ on the transmitting transducer. (**b**) Signals $S_{R}\left(t\right)$ on the receiving transducer. (**c**) Signals $S_{LMS}\left(t\right)$ after LMS adaptive filtering. (**d**) Signals $S_{RLS}\left(t\right)$ after RLS adaptive filtering. (**e**) Signals $S_{PNLMS}\left(t\right)$ after PNLMS adaptive filtering. (**f**) Signals $S_{DPRNN}\left(t\right)$ after DPRNN treating.

**Figure 12 sensors-24-02141-f012:**
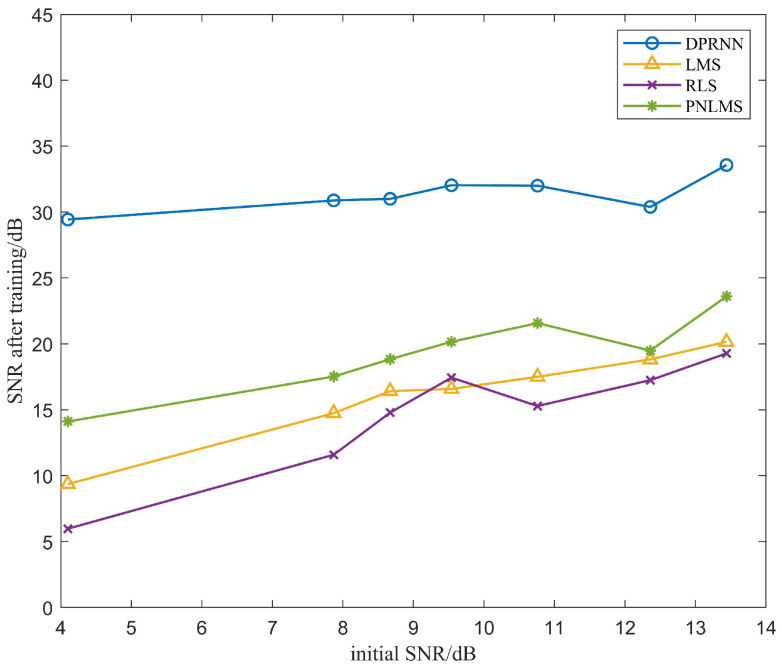
Effect of initial received signals’ SNR on SNR of signals treated with LMS, RLS, PNLMS and DPRNN, respectively.

**Figure 13 sensors-24-02141-f013:**
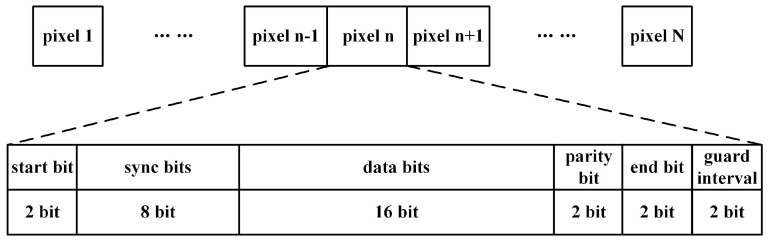
Data structure of Pixel cell.

**Figure 14 sensors-24-02141-f014:**
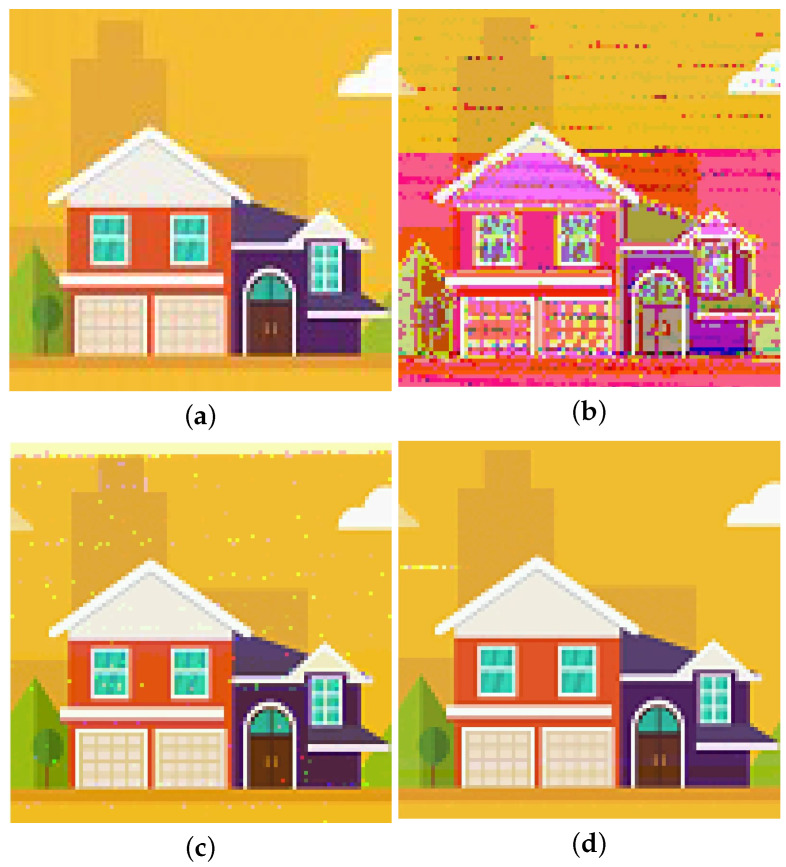
Images recovered from signals treated with different approaches of echo cancellation. (**a**) image collected by FPGA chip in the transmitting side. (**b**) image recovered from signals without any echo cancellation. (**c**) image recovered from signals with LMS adaptive filtering. (**d**) image recovered from signals with DPRNNN-based echo cancellation.

**Table 1 sensors-24-02141-t001:** SNRs of received signals $S_{R}\left(t\right)$ under different channel conditions.

Channel Length/mm	30	40	50	50	50	50	60
Offset Distance/mm	0	0	0	6	12	18	0
SNR/dB	12.36	10.76	13.44	9.54	7.87	4.10	8.67

**Table 2 sensors-24-02141-t002:** Signal quality with different methods of echo cancellation.

Evaluation Index	$S_{\mathrm{LMS}}\left(t\right)$	$S_{\mathrm{PNLMS}}\left(t\right)$	$S_{\mathrm{RLS}}\left(t\right)$	$S_{\mathrm{DPRNN}}\left(t\right)$
SNR (dB)	9.331	14.112	5.974	29.437
SISNR (dB)	22.911	43.688	16.243	113.826
NCC	0.7533	0.8952	0.6391	0.9829
BER	5.593 × $10^{-2}$	2.182 × $10^{-3}$	3.742 × $10^{-2}$	3.502 × $10^{-4}$
ERLE	6.47	17.82	4.22	32.53

**Table 3 sensors-24-02141-t003:** Evaluation of image restoration using signals with different echo cancellation methods.

Evaluation Index	Image without Echo Cancellation	Image with LMS Filter	Image with DPRNN Model
PSNR/dB	11.8911	19.1776	24.8396
SSIM	0.4144	0.8911	0.9471
ΔE	17.1429	5.6638	3.6256

## Data Availability

All data included in this study are available upon request by contact with the corresponding author.

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
