# Peer review of "Ultrasonic Through-Metal Communication Based on Deep-Learning-Assisted Echo Cancellation"

_sensors, 2024, doi:10.3390/s24072141_

Round 1
Reviewer 1 Report
Comments and Suggestions for Authors
A deep learning method, namely DPRNN, is introduced to solve the echo cancellation problem. Extensive simulations verify the effectiveness of the proposed approach. Overall of this paper is well-written and the results are interesting. When revising, the following can be considered.
1) The concept of echo cancellation by using deep learning method is not new. As far as I know, most of such methods are extended the deep learning algorithms to acoustics echo cancellation. The following references should be mentioned and described.
[a]Nonlinear acoustic echo cancellation with deep learning(2021)arXiv
[b]A robust and cascaded acoustic echo cancellation based on deep learning(2020)
2) In Introduction, the authors introduce some adaptive filtering algorithms for echo cancellation, such as LMS, NLMS, and RLS. Also, in simulations, the authors only compared to the classical LMS algorithm. For echo cancellation, the subband filter and sparse filter also should be mentioned. For example.
Subband filter
Proposing a robust RLS based subband adaptive filtering for audio noise cancellation(2024)APAC
Proximal normalized subband adaptive filtering for acoustic echo cancellation
…
Sparse filter
An adjusting-block based convex combination algorithm for identifying block-sparse system(2017)SP
Convergence analysis of zero attracting natural gradient non-parametric maximum likelihood algorithm(2018)TCAS
…
If possible, I also suggest the authors compare the DPRNN algorithm with one of the above-metioned algorithms, since the LMS algorithm is old and it is easy to exceed the performance of LMS.
3) The ERLE can be used to evaluate the performance.
Comments on the Quality of English Language
English presentation can be further improved.
Author Response
Thank you very much for taking the time to review this manuscript. We appreciate your valuable feedback and constructive comments. We have carefully considered your suggestions and make some changes.
Please see the attachment for the corresponding response.

Reviewer 2 Report
Comments and Suggestions for Authors
1.Channel Characterization: The study needs a more comprehensive analysis of the ultrasonic channel beyond impulse and magnitude responses. Consider including multipath effects and signal attenuation.
2.Model Training and Validation: Provide more details on the DPRNN model's training and validation process. Explain the rationale behind choosing model parameters and training epochs.
3.Image Quality Analysis: Add quantitative analysis for the quality of images recovered using different echo cancellation methods. Compare details like distortion levels and color fidelity.
4.Data and Statistical Analysis: The paper should present more detailed data and statistical analysis to support the conclusions, especially regarding the DPRNN's performance over the LMS method.
Comments on the Quality of English Language1.Sentence Structure: Some sentences could be simplified for better readability.
2.Consistency in Terms: Ensuring consistent use of technical terms throughout the paper would enhance its professional quality.
Author Response

(The authors gave the same response as above.)

Reviewer 3 Report
Comments and Suggestions for Authors
The presented work addresses ultrasonic through-metal communications, which are crucial for various applications such as monitoring the internal status of a nuclear reactor and gathering data from sensors deployed in high-pressure pipes. Ultrasound is the sole viable option for communication through metal shields since electromagnetic waves are blocked by the Faraday shielding effect. The authors propose the use of a dual-path recurrent neural network (DPRNN) deep learning algorithm for echo cancellation in an ultrasonic through-metal communication system. They develop an experimental setup and convincingly demonstrate that the DPRNN algorithm leads to significant improvements in terms of scale-invariant signal-to-noise ratio (SISNR), normalized correlation coefficient (NCC), and bit error rate (BER). The paper provides a thorough comparison with conventional Least Mean Square (LMS) adaptive filtering, showing its relatively lower efficiency compared to the proposed DPRNN method.
The manuscript is well-written and suitable for publication with a few minor corrections:
· Line 68: “the transmission of an image with 100×100 pitches was achieved”. Possibly, the word “pixels” was intended.
· Line 75: The term “Sandwich structure” may not be appropriate for the configuration presented in Figure 1, as the latter is traditionally referred to as “two thin sheets of high-performing material filled between them with a low-density material” (An introduction to sandwich construction. D. Zenkert, H.G. Allen - 1995).
· Figure 6: lacks a caption.
Author Response

(The authors gave the same response as above.)

Round 2
Reviewer 1 Report
Comments and Suggestions for Authors
no comments